# `FlashSchNet`: Fast and Accurate Coarse-Grained Neural Network Molecular Dynamics

Pingzhi Li [1]   Hongxuan Li [1 2]   Zirui Liu [3]   Xingcheng Lin [2]   Tianlong Chen [1]

## Abstract

Graph neural network (GNN) potentials such as SchNet improve the accuracy and transferability of molecular dynamics (MD) simulation by learning many-body interactions, but remain slower than classical force fields due to fragmented kernels and memory-bound pipelines that underutilize GPUs. We show that a missing principle is making GNN-MD *IO-aware*, carefully accounting for reads and writes between GPU high-bandwidth memory (HBM) and on-chip SRAM. We present `FlashSchNet`, an efficient and accurate IO-aware SchNet-style GNN-MD framework built on four techniques: (1) *flash radial basis*, which fuses pairwise distance computation, Gaussian basis expansion, and cosine envelope into a single tiled pass, computing each distance once and reusing it across all basis functions; (2) *flash message passing*, which fuses cutoff, neighbor gather, filter multiplication, and reduction to avoid materializing edge tensors in HBM; (3) *flash aggregation*, which reformulates scatter-add via CSR segment reduce, reducing atomic writes by a factor of feature dimension and enabling contention-free accumulation in both forward and backward passes; (4) *channel-wise 16-bit quantization* that exploits the low per-channel dynamic range in SchNet MLP weights to further improve throughput with negligible accuracy loss. On a single NVIDIA RTX PRO 6000, `FlashSchNet` achieves **1000 ns/day** aggregate simulation throughput over 64 parallel replicas on coarse-grained (CG) protein containing 269 beads (**6.5×** faster than CGSchNet baseline with **80% reduction** of peak memory), surpassing classical force fields (*e.g.* MARTINI) while retaining SchNet-level accuracy and transferability.

---

[1]The University of North Carolina at Chapel Hill [2]North Carolina State University [3]University of Minnesota at Twin Cities. Correspondence to: Pingzhi Li <pingzhi@cs.unc.edu>.

*Proceedings of the 43rd International Conference on Machine Learning*, Seoul, South Korea. PMLR 306, 2026. Copyright 2026 by the author(s).

## 1. Introduction

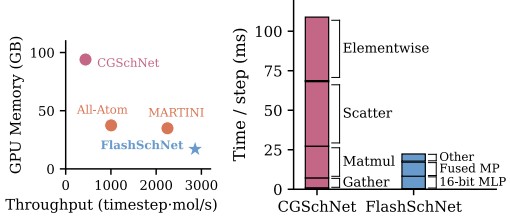

*Figure 1.* Left: Memory-throughput trade-off for SchNet-style GNN-MD. `FlashSchNet` achieves 5× memory reduction while improving throughput by 6× over CGSchNet baseline. Right: Step time breakdown showing `FlashSchNet` eliminates scatter and element-wise bottlenecks via fused kernels and 16-bit quantization. All are evaluated on a 269-bead protein (1ENH) with 64 replicas.

Molecular dynamics (MD) simulation is a core tool in computational chemistry, drug discovery, and materials science, offering a computational microscope for probing molecular motion at atomic resolution (Karplus & McCammon, 2002; Dror et al., 2012; Hollingsworth & Dror, 2018). By numerically integrating Newton's equations of motion, MD produces time-resolved trajectories that connect microscopic interactions to macroscopic observables, enabling thermodynamic estimation, conformational exploration, and mechanistic study of rare events (Chodera & Noé, 2014). In practice, however, classical MD faces a persistent trade-off: empirical force fields are fast but approximate, while first-principles MD such as Car–Parrinello is more faithful but orders of magnitude more expensive (Car & Parrinello, 1985). Even when adopting widely used coarse-grained models such as MARTINI (Marrink et al., 2007), which sacrifice atomistic detail for efficiency, the repeated evaluation of forces over millions to billions of timesteps remains a fundamental bottleneck, limiting accessible timescales and system sizes in routine workflows (De Vivo et al., 2016).

Motivated by this gap, graph neural network (GNN) potentials have rapidly emerged as a leading class of machine-learned force fields (MLFFs). Rooted in geometric deep learning principles (Bronstein et al., 2021), these models represent atoms as nodes and local interactions as edges, and use message passing to capture many-body effects in

a data-driven yet physically structured manner (Gilmer et al., 2017). SchNet (Schütt et al., 2017) and subsequent geometric GNNs (for example, DimeNet (Gasteiger et al., 2020) and $E(3)$-equivariant architectures such as NequIP (Batzner et al., 2022), Allegro (Musaelian et al., 2023), and MACE (Batatia et al., 2022)) have demonstrated strong accuracy and improved transferability across chemical environments, bringing ML potentials closer to first-principles fidelity at a fraction of the compute.

Yet higher accuracy has not translated into faster wall-clock simulation. In SchNet-style GNN-MD, continuous-filter convolution (CFConv) repeatedly constructs edge-wise features (distances, radial bases, cutoffs) and applies small MLPs, followed by scatter-style aggregation over dynamic neighborhoods. When implemented with high-level deep learning frameworks such as PyTorch and JAX, this computation fragments into many kernels and repeatedly materializes intermediate edge tensors in GPU high-bandwidth memory (HBM), while aggregation suffers from heavy synchronization due to contended atomic updates. As a result, the workload is strongly memory-bound and under-utilizes the GPU despite modest nominal FLOPs. For example, CGSchNet (Charron et al., 2025) running 64 parallel replicas on a coarse-grained (CG) protein with 269 beads achieves only **2.5%** model FLOPs utilization (MFU)[1]. These observations point to a missing principle for practical GNN-MD: *making the pipeline IO-aware by optimizing reads and writes between HBM and on-chip SRAM.*

We identify four major bottlenecks in SchNet-style GNN-MD, which all stem from memory IO: ❶ *Radial basis expansion* computes pairwise distances, Gaussian basis values, and cosine cutoffs in separate kernels, materializing intermediate tensors (distances, expanded bases, cutoff values) to HBM even though each is consumed only once; ❷ *Message passing* launches distinct operations for cutoff masking, neighbor gather, filter multiplication, and scatter aggregation, writing large edge tensors of size $O(E \times F)$ (number of edges $\times$ feature dimension) to HBM between stages; ❸ *Scatter aggregation* uses atomic additions to accumulate messages, incurring $O(E \times F)$ conflict atomic writes that serialize under high neighborhood density; ❹ *Filter networks* repeatedly load MLP weights for every edge, making these small matrix multiplications strongly bandwidth-bound.

❶ *Flash radial basis* fuses pairwise distance computation, Gaussian basis expansion, and cosine envelope into a single tiled pass, computing each distance once and reusing it on-chip across all basis functions. ❷ *Flash message passing* fuses cutoff masking, neighbor gather, filter multiplication, and reduction into one kernel, eliminating materialization

of intermediate edge tensors in HBM. ❸ *Flash aggregation* reformulates scatter-add via CSR segment reduce, reducing atomic writes by a factor of feature dimension and enabling contention-free accumulation in both forward and backward passes. ❹ *Channel-wise 16-bit quantization* exploits low per-channel dynamic range in SchNet MLP weights to further improve throughput with negligible accuracy loss. As demonstrated in Figure 1, FlashSchNet achieves significant end-to-end speedup and memory savings. Our contributions are summarized as:

- We identify IO inefficiency as the key bottleneck in SchNet-style GNN-MD and show how to exploit inherent model structure, *i.e.*, graph sparsity for contention-free CSR aggregation and low per-channel dynamic range for 16-bit weight quantization, to reduce memory traffic at the algorithmic level.

- We translate these insights into four kernel-level implementations (*i.e.*, flash radial basis, flash message passing, flash aggregation, and quantized filter networks) that together eliminate intermediate tensor materialization, deliver end-to-end speedup and memory saving.

- We combine these techniques into FlashSchNet, achieving **6.5× speedup** and **80% memory reduction** over CGSchNet baseline. To our knowledge, this is the first SchNet-style GNN-MD that surpasses classical coarse-grained force fields such as MARTINI, reaching **1000 ns/day** aggregate simulation throughput over 64 parallel replicas on coarse-grained protein containing 269 beads on a single RTX PRO 6000, while retaining the accuracy and transferability of learned potentials.

## 2. Related work

**Molecular dynamics simulation and machine-learned force fields.** Molecular dynamics (MD) simulation is fundamental for studying molecular systems. Traditional force fields (e.g., AMBER (Wang et al., 2004), CHARMM (Brooks et al., 2009)) are computationally efficient but limited in transferability due to fixed functional forms. *Ab initio* MD (Car & Parrinello, 1985; Kühne et al., 2020) achieves high accuracy via first-principles calculations, but its $\mathcal{O}(N^3)$ scaling restricts applicability. Machine-learned force fields (MLFFs) bridge this accuracy-efficiency gap. Early approaches include kernel methods, *e.g.* GAP (Bartók et al., 2010) and sGDML (Chmiela et al., 2019), and neural network potentials (Behler & Parrinello, 2007). GNN-based MLFFs such as SchNet (Schütt et al., 2017), DimeNet (Gasteiger et al., 2020), and PhysNet (Unke & Meuwly, 2019) operate directly on molecular graphs with improved generalization. $E(3)$-equivariant models including NequIP (Batzner et al., 2022), Allegro (Musaelian et al., 2023), and MACE (Batatia et al., 2022) achieve superior

---

[1]MFU is computed as achieved TFLOPs/s divided by the GPU peak TFLOPs/s. We enable TF32 tensor cores and report measurements on a single NVIDIA RTX PRO 6000.

data efficiency by preserving geometric symmetries, though at higher computational cost. Recent universal MLFFs (Ju et al., 2025; Neumann et al., 2024; Yang et al., 2024) enhance transferability through large-scale training, but inference cost remains a bottleneck for large-scale simulations.

Recent works have explored coarse-grained GNN force fields to improve scalability while maintaining physical fidelity. Airas and Zhang (Airas & Zhang, 2026) introduce a solvent-aware CG potential by distilling structural priors from protein language models, focusing on secondary structure and solvent exposure. Majewski et al. (Majewski et al., 2023) develop neural CG force fields that reproduce protein thermodynamics across multiple proteins using long atomistic trajectories. Charron et al. (Charron et al., 2025) propose a transferable CG model that generalizes to unseen sequences and accurately predicts folding landscapes and mutation effects. While these models improve physical fidelity and, in some cases, generalization, their inference remains memory- and IO-bound, limiting scalability to long trajectories or large biomolecular systems.

**Efficient graph neural networks.** A line of work improves GNN efficiency by optimizing sparse message-passing operators at various system levels, including graph-centric frameworks with dedicated CUDA kernels (Wang et al., 2019; Fey & Lenssen, 2019), runtime systems that adapt execution to graph structure (Wang et al., 2021), compiler stacks that fuse operators and reduce kernel launches (Xie et al., 2022), and accelerated sparse primitives such as SpMM and SDDMM with CSR-compatible designs (Chen et al., 2020; Huang et al., 2020; Rahman et al., 2021). These efforts primarily target generic GNN workloads on large, mostly static graphs where sparse linear algebra dominates. SchNet-style GNN-MD differs significantly because dynamic neighbor lists, continuous-filter convolutions with per-edge MLPs, and the need for efficient backward passes for force computation make repeated edge-tensor materialization and contention-heavy scatter-add the key bottlenecks, motivating our IO-aware fusion and contention-free CSR-style aggregation.

**Memory-bound runtime optimization.** Modern GNN and ML potential workloads are often memory-bound, as irregular gather/scatter interleaved with small dense kernels makes throughput dominated by data movement rather than FLOPs. FlashAttention (Dao et al., 2022) exemplifies IO-aware algorithm design that explicitly reasons about HBM to SRAM traffic and uses tiling and recomputation to maximize on-chip reuse. In GNN runtimes, high-level frameworks often execute message construction and aggregation as fragmented kernels that materialize intermediates and suffer from atomic contention (Gong et al., 2025), prompting fusion techniques that reduce memory traffic and

*Table 1.* Summary of notation used throughout the paper.

| Symbol | Description |
|---|---|
| $N$ | Number of atoms or beads |
| $E$ | Number of directed edges in the neighbor graph |
| $D$ | Hidden feature dimension |
| $D_r$ | Radial basis dimension |
| $T$ | Number of interaction blocks |
| $r_{\text{cut}}$ | Cutoff radius for neighbor list construction |
| $\mathbf{r}_i \in \mathbb{R}^3$ | Position of atom $i$ |
| $\mathbf{x}_i^{(t)} \in \mathbb{R}^D$ | Hidden feature of atom $i$ at layer $t$ |
| $\mathbf{X}^{(t)} \in \mathbb{R}^{N \times D}$ | Stacked hidden features over all atoms |
| $\texttt{src}, \texttt{dst} \in \{1, \dots, N\}^E$ | Source and destination index arrays for edges |
| $d_e \in \mathbb{R}$ | Scalar distance for edge $e$ |
| $\mathbf{b}_e \in \mathbb{R}^{D_r}$ | Radial basis vector for edge $e$ |
| $\mathbf{B} \in \mathbb{R}^{E \times D_r}$ | Stacked radial basis over all edges |
| $\mathbf{w}_e \in \mathbb{R}^D$ | Continuous filter for edge $e$ |
| $\mathbf{W} \in \mathbb{R}^{E \times D}$ | Stacked filters over all edges |
| $\mathbf{m}_e^{(t)} \in \mathbb{R}^D$ | Message for edge $e$ at layer $t$ |
| $\mathcal{E}$ | Total potential energy |
| $\epsilon_i \in \mathbb{R}$ | Per-atom energy contribution from atom $i$ |
| $\mathbf{F}_i \in \mathbb{R}^3$ | Force on atom $i$ |

launch overhead (Liu et al., 2024). For molecular simulation, TorchMD-Net 2.0 achieves substantial speedups by engineering the simulation stack with optimized neighbor search and efficient force evaluation (Pelaez et al., 2024). These efforts highlight that large wall-clock gains require end-to-end, IO-aware redesign that co-optimizes feature construction, message passing, and aggregation rather than accelerating isolated kernels.

## 3. Background

We summarize our used notation in Table 1. Section 3.1 reviews molecular dynamics and force evaluation. Section 3.2 presents the SchNet architecture, highlighting the operators that dominate runtime. Section 3.3 demonstrates the hardware bottlenecks that motivate our IO-aware design.

### 3.1. Molecular dynamics and force evaluation

Molecular dynamics (MD) simulates the evolution of atom/bead positions $\{\mathbf{r}_i\}_{i=1}^N$ by repeatedly evaluating forces $\{\mathbf{F}_i\}$ and integrating the equations of motion (Karplus & McCammon, 2002). In energy-based MD, a force field defines a scalar potential energy $\mathcal{E}(\{\mathbf{r}_i\})$, and forces are:

$$\mathbf{F}_i = -\nabla_{\mathbf{r}_i} \mathcal{E} \in \mathbb{R}^3.$$

A time integrator then updates the state via:

$$\mathbf{r}_i \leftarrow \mathbf{r}_i + \Delta t\, \mathbf{v}_i + \cdots, \qquad \mathbf{v}_i \leftarrow \mathbf{v}_i + \Delta t\, \mathbf{F}_i / m_i + \cdots,$$

where $\mathbf{v}_i$ is the velocity, $m_i$ is the mass, and the omitted terms depend on the chosen thermostat integrator (*e.g.*, Langevin). Each MD step therefore requires (i) evaluating $\mathcal{E}$ and (ii) backpropagating to obtain $\mathbf{F}_i$, making the end-to-end throughput dominated by both *forward* and *backward* cost of the learned potential.

## 3.2. SchNet model

SchNet (Schütt et al., 2017) is a continuous-filter message-passing network that predicts potential energy $E$ from atom positions $\{\mathbf{r}_i\}_{i=1}^{N}$ and types $\{Z_i\}_{i=1}^{N}$, and obtains forces via $\mathbf{F}_i = -\nabla_{\mathbf{r}_i} E$. The model maintains atom-wise hidden features $\mathbf{x}_i^{(t)} \in \mathbb{R}^D$ (stacked as $\mathbf{X}^{(t)} \in \mathbb{R}^{N \times D}$) and iteratively applies distance-dependent interactions over a sparse neighbor graph induced by a radial cutoff. We now describe the computational pipeline and dominating operators, following the specific architecture used in Charron et al. (2025).

**Building the neighbor list.** Given positions, SchNet first constructs a neighbor list with cutoff radius $r_{\text{cut}}$, represented as two index arrays $\mathtt{src}, \mathtt{dst} \in \{1, \ldots, N\}^E$ indexing the $E$ directed edges. Each edge $e$ encodes an interaction from source $j = \mathtt{src}[e]$ to destination $i = \mathtt{dst}[e]$.

**Distances and radial basis.** For each edge $e$, SchNet computes the displacement vector and scalar distance

$$\mathbf{u}_e = \mathbf{r}_{\mathtt{dst}[e]} - \mathbf{r}_{\mathtt{src}[e]} \in \mathbb{R}^3, \qquad d_e = \|\mathbf{u}_e\|_2,$$

and expands $d_e$ into a $D_r$-dimensional radial basis vector $\mathbf{b}_e = \text{RBF}(d_e) \in \mathbb{R}^{D_r}$, typically modulated by a smooth cutoff envelope. Stacking over all edges yields $\mathbf{B} \in \mathbb{R}^{E \times D_r}$.

**Filter network.** A small MLP maps each radial basis vector to a $D$-dimensional continuous filter:

$$\mathbf{w}_e = \text{MLP}_{\text{filter}}(\mathbf{b}_e) \in \mathbb{R}^D,$$

producing stacked filters $\mathbf{W} \in \mathbb{R}^{E \times D}$. Because this MLP is evaluated per edge, the resulting tensor scales as $O(E \times D)$ and constitutes a major source of memory traffic.

**CFConv message passing and aggregation.** The continuous-filter convolution (CFConv) forms edge messages by element-wise multiplication of the source feature with the learned filter, $\mathbf{m}_e^{(t)} = \mathbf{x}_{\mathtt{src}[e]}^{(t)} \odot \mathbf{w}_e \in \mathbb{R}^D$, and aggregates them onto destination nodes via a sum over incoming edges: $\mathbf{h}_i^{(t)} = \sum_{e: \mathtt{dst}[e]=i} \mathbf{m}_e^{(t)} \in \mathbb{R}^D$. A point-wise update network with residual connection then produces $\mathbf{x}_i^{(t+1)}$; this interaction block is repeated $T$ times. Standard implementations realize aggregation via $\mathtt{scatter\_add}$, which incurs $O(E \times D)$ atomic writes with significant contention when multiple edges share the same destination.

**Energy readout.** After $T$ interaction blocks, an output MLP maps atom features to per-atom energy contributions:

$$\epsilon_i = \text{MLP}_{\text{out}}(\mathbf{x}_i^{(T)}) \in \mathbb{R}, \qquad E = \sum_{i=1}^{N} \epsilon_i,$$

optionally combined with prior energy terms (*e.g.*, bonded interactions) as $E \leftarrow E + E_{\text{prior}}$ (Charron et al., 2025).

SchNet contains multiple MLP submodules, *i.e.* $\text{MLP}_{\text{filter}}$, block-wise update networks, and $\text{MLP}_{\text{out}}$. All of them are bandwidth-bound due to repeated weight loading.

**Forces via autodiff.** For molecular dynamics, forces are obtained by differentiating the scalar energy with respect to positions: $\mathbf{F}_i = -\nabla_{\mathbf{r}_i} E \in \mathbb{R}^3$. This requires backpropagating through neighbor-list-indexed distance and RBF computations, all MLP submodules, and the aggregation operator. Crucially, the backward pass through aggregation also involves scatter-style accumulation (now over source nodes), making both forward and backward efficiency essential for practical MD throughput.

## 3.3. Challenges of hardware performance

We focus on GPU. Modern GPUs offer high peak FLOPs but are frequently limited by memory traffic between high-bandwidth memory (HBM) and on-chip storage. For SchNet-style GNN-MD, the dominant operators are sparse, index-based pipelines over the neighbor graph, as they repeatedly materialize large edge tensors (*e.g.*, $\mathbf{B} \in \mathbb{R}^{E \times D_r}$ and $\mathbf{W} \in \mathbb{R}^{E \times D}$) and perform scatter-style reductions whose arithmetic intensity is low relative to their HBM read/write volume. This makes runtime primarily *bandwidth-bound*, and fragmented kernels further reduce effective throughput by repeatedly loading and storing intermediates.

**HBM traffic from edge intermediates.** The core SchNet pipeline expands edge distances into $\mathbf{B} \in \mathbb{R}^{E \times D_r}$ and filters into $\mathbf{W} \in \mathbb{R}^{E \times D}$, and conceptually forms edge messages $\mathbf{M}^{(t)} \in \mathbb{R}^{E \times D}$ with rows $\mathbf{m}_e^{(t)}$. Even when compute per element is modest, writing and rereading these edge tensors incurs $O(E \cdot D_r)$ and $O(E \cdot D)$ HBM traffic per block, which is amplified across $T$ interaction blocks and again in the backward pass required for force evaluation.

**Scatter contention in graph reductions.** Aggregation in CFConv typically uses $\mathtt{scatter\_add}$ to accumulate $\mathbf{m}_e^{(t)}$ into destination nodes $\mathbf{h}_i^{(t)}$. This performs $O(E \cdot D)$ atomic updates, and when many edges share the same destination (*i.e.*, high local degree), concurrent atomics serialize and significantly lower throughput. Moreover, the backward pass through aggregation also requires scatter accumulation, so contention impacts both forward and backward passes, directly limiting MD wall-clock step time.

**Mixed precision and Tensor Cores.** GPUs provide specialized Tensor Cores that accelerate matrix-multiply and fused MLP primitives at various precisions (*e.g.* FP16). In SchNet, the filter, update, and readout networks are composed of MLPs whose weights are repeatedly loaded, making them sensitive to both compute throughput and memory bandwidth. Using FP16 weights and activations, while keep-

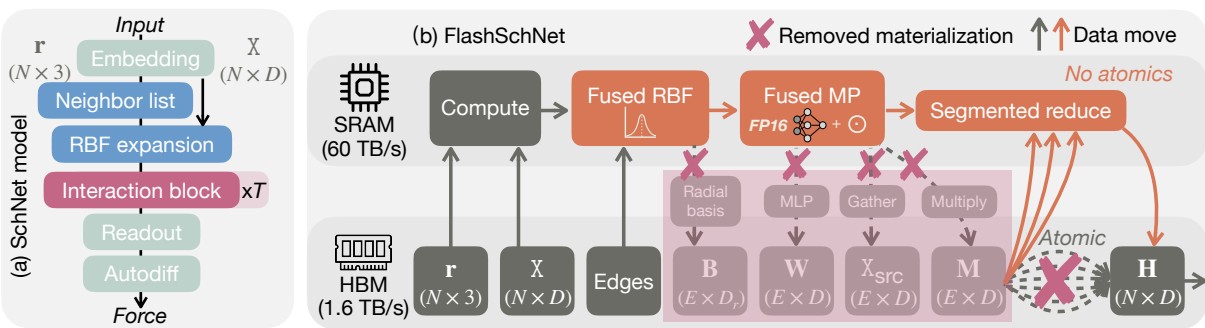

*Figure 2.* (a) SchNet model architecture for molecular dynamics: atom positions **r** and embeddings **X** are processed through neighbor list construction, radial basis expansion, and $T$ interaction blocks, followed by energy readout and autodiff for force computation. (b) FlashSchNet IO-aware execution model. The baseline pipeline (bottom, shaded) materializes intermediate edge tensors ($\mathbf{B} \in \mathbb{R}^{E \times D_r}$, $\mathbf{W}, \mathbf{X}_{\mathrm{src}}, \mathbf{M} \in \mathbb{R}^{E \times D}$) to HBM and uses atomic scatter for aggregation. FlashSchNet (top, orange) fuses these operations into three kernels that keep intermediates in SRAM: *Fused RBF* computes distances, Gaussian basis expansion, and cosine envelope in one pass; *Fused MP* combines FP16 filter MLP, neighbor gather, and element-wise multiplication; *Segmented reduce* replaces atomic scatter-add with contention-free CSR-style accumulation. Red crosses indicate eliminated HBM materializations. The FlashSchNet pipeline reduces memory traffic by $\sim E/N$ and removes all atomic contention.

ing key accumulations and force outputs in FP32, can reduce weight/activation traffic and increase compute throughput by mapping these MLPs onto Tensor Cores.

# 4. Flash-SchNet

This section presents FlashSchNet, an IO-aware SchNet-style GNN-MD implementation that accelerates the *end-to-end* MD step, including forward energy evaluation + backward force computation. As shown in Figure 2, our design targets the dominant bottlenecks identified in Section 3. At a high level, FlashSchNet follows the baseline CGSchNet (Charron et al., 2025) architecture, but (a) fuses single-use edge pipelines to avoid materializing large intermediates in HBM, (b) replaces atomic scatter reductions with contention-free CSR segment reductions, and (c) applies channel-wise 16-bit quantization to MLP submodules to reduce both compute time and memory traffic losslessly.

**Computation targeted by FlashSchNet.** Consider one interaction block at layer index $t$ under neighbor graph $(\mathrm{src}, \mathrm{dst}) \in \{1, \ldots, N\}^E$. For edge $e$, define displacement $\mathbf{u}_e = \mathbf{r}_{\mathrm{dst}[e]} - \mathbf{r}_{\mathrm{src}[e]}$, distance $d_e = \|\mathbf{u}_e\|_2$, radial basis $\mathbf{b}_e = \mathrm{RBF}(d_e) \in \mathbb{R}^{D_r}$, and a smooth cutoff envelope $C(d_e) \in \mathbb{R}$. The CFConv aggregation can be written as

$$\mathbf{h}_i^{(t)} = \sum_{e:\,\mathrm{dst}[e]=i} \left( \mathbf{x}_{\mathrm{src}[e]}^{(t)} \odot \mathbf{w}_e \right),$$

where $\mathbf{w}_e = \mathrm{MLP}_{\mathrm{filter}}(\mathbf{b}_e \cdot C(d_e)) \in \mathbb{R}^D$. Baseline implementations typically materialize $\mathbf{B} \in \mathbb{R}^{E \times D_r}$ and $\mathbf{W} \in \mathbb{R}^{E \times D}$ as HBM intermediates, and realize the sum using scatter_add, leading to large memory traffic and atomic contention. FlashSchNet computes the same $\mathbf{h}_i^{(t)}$ while avoiding edge tensor materialization and eliminating

atomics on the aggregation path.

## 4.1. IO-aware reformulation of SchNet interaction

**Single-use edge pipeline.** The per-edge computation forms a single-use chain distance to radial-basis to filter MLP to gated message. We treat this chain as a streaming operator and fuse it so that $\mathbf{u}_e$, $d_e$, $\mathbf{b}_e$, and intermediate activations inside $\mathrm{MLP}_{\mathrm{filter}}$ are produced and consumed on chip. Conceptually, we replace explicit edge tensors with a fused edge operator $\mathbf{h}_i^{(t)} = \sum_{e:\,\mathrm{dst}[e]=i} \Psi\big(\mathbf{x}_{\mathrm{src}[e]}^{(t)}, \mathbf{r}_{\mathrm{src}[e]}, \mathbf{r}_{\mathrm{dst}[e]}\big)$, where $\Psi$ encapsulates distance computation, radial basis and envelope evaluation, filter MLP, and gating.

**Precision contract for force-based simulation.** Forces require gradients through $d_e = \|\mathbf{u}_e\|_2$ and the cutoff and basis functions. We keep positions $\mathbf{r}_i$, distances $d_e$, energy accumulation $\mathcal{E}$, and force outputs $\mathbf{F}_i$ in FP32, and use FP32 accumulation for reductions. W16A16 is applied to SchNet MLP submodules, as described in Section 4.4.

## 4.2. Flash message passing fused edge computation

**Fused forward operator.** For each edge $e$ with $(j, i) = (\mathrm{src}[e], \mathrm{dst}[e])$, we compute:

$$\mathbf{u}_e = \mathbf{r}_i - \mathbf{r}_j, \quad d_e = \|\mathbf{u}_e\|_2,$$
$$\tilde{\mathbf{b}}_e = \mathrm{RBF}(d_e) \cdot C(d_e), \quad \mathbf{w}_e = \mathrm{MLP}_{\mathrm{filter}}(\tilde{\mathbf{b}}_e),$$

and then form the message $\mathbf{m}_e^{(t)} = \mathbf{x}_j^{(t)} \odot \mathbf{w}_e$ and directly feed it into aggregation for $\mathbf{h}_i^{(t)}$. This removes the need to materialize B and W as HBM intermediates.

**On-chip reuse.** We tile edges and organize computation so that values reused within a short window, such as $\mathbf{r}_i$ for edges sharing the same destination, are kept in registers or shared memory. This reduces global memory traffic even when it introduces modest recomputation.

### 4.3. Flash aggregation segmented reductions

**Destination grouped segmented reduction in forward pass.** To avoid atomic contention, we reorder edges by destination and perform a segmented reduction. Let $\texttt{dst\_ptr} \in \{0, \ldots, E\}^{N+1}$ and $\texttt{perm} \in \{1, \ldots, E\}^E$ define a destination grouped layout where edges for node $i$ occupy as

$$p \in [\texttt{dst\_ptr}[i], \texttt{dst\_ptr}[i+1]).$$

Then

$$\mathbf{h}_i^{(t)} = \sum_{p=\texttt{dst\_ptr}[i]}^{\texttt{dst\_ptr}[i+1]-1} \mathbf{m}_{\texttt{perm}[p]}^{(t)}.$$

We assign exclusive ownership of each destination segment to one block, accumulate in registers, and write once per feature channel.

**Source grouped segmented reduction in backward pass.** The dominant gradient with respect to source features is as:

$$\nabla \mathbf{x}_j^{(t)} = \sum_{e:\ \texttt{src}[e]=j} \nabla \mathbf{h}_{\texttt{dst}[e]}^{(t)} \odot \mathbf{w}_e.$$

We avoid atomic contention by building a source grouped layout and applying the same exclusive ownership principle to accumulate $\nabla \mathbf{x}_j^{(t)}$.

**Index construction under dynamic neighbor lists.** Neighbor lists may change across MD steps, so the grouped layouts must be rebuilt when $(\texttt{src}, \texttt{dst})$ changes. We construct the destination grouped and source grouped indices using bucket sort on $\texttt{dst}$ and $\texttt{src}$, respectively, producing contiguous edge segments per node that enable exclusive ownership segmented reductions. We provide isolated profiling of this bucket sort overhead in Appendix C.

### 4.4. W16A16 mixed precision for MLP submodules

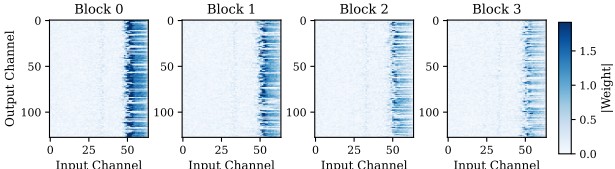

*Figure 3.* Filter networks show clear channel-wise magnitude distribution, motivating channel quantization for lossless acceleration.

**Motivation.** SchNet filter networks demonstrate a clear channel-wise magnitude structure. As shown in Figure 3, weight magnitudes concentrate unevenly across output channels, and this pattern is consistent across interaction blocks, motivating channel-wise quantization as a near-lossless way to reduce MLP computing and IO cost.

**Channel-wise quantization.** We apply W16A16 (16-bit weight, 16-bit activation) to all MLP submodules, including $\text{MLP}_{\text{filter}}$, blockwise update networks, and the readout network $\text{MLP}_{\text{out}}$. We adapt Optimal Brain Compression (Frantar et al., 2023) to compute per-channel quantization scales for each linear layer, minimizing the quantization loss. With FP16 weights and intermediate activations, MLP GEMMs map to Tensor Cores and reduce weight and activation traffic over FP32. We keep positions $\mathbf{r}_i$, distances $d_e$, energy accumulation $\mathcal{E}$, and force outputs $\mathbf{F}_i$ in FP32, and use FP32 accumulation for reductions as described in Section 4.1.

### 4.5. End-to-end integration

At each MD step, we build the neighbor list, update segmented reduction indices when enabled, run $T$ interaction blocks with fused message passing and segmented reductions, compute energy via the readout network, and obtain forces by autodiff as $\mathbf{F}_i = -\nabla_{\mathbf{r}_i}\mathcal{E}$. The configuration enables controlled ablations of fusion, segmented reductions, and W16A16 in Section 5.

**Cost analysis.** FlashSchNet avoids materializing $\mathtt{B} \in \mathbb{R}^{E \times D_r}$ and $\mathtt{W} \in \mathbb{R}^{E \times D}$ in HBM, reducing dominant IO per step from $\text{IO}^{\texttt{base}} = \Theta\big(T \cdot E(D_r + D)\big) + \Theta\big(T \cdot ED\big)$ to $\text{IO}^{\texttt{flash}} = \Theta\big(T \cdot (ED + ND)\big)$, eliminating radial-basis and filter materialization, and replacing $O(ED)$ contention-heavy atomic aggregation with $O(ND)$ contention-free segment stores. Since $E \gg N$ in typical GNN-MD (*e.g.*, $10^5$ vs. $10^2$), total IO drops by $\sim E/N$. 16-bit quantization further reduces MLP weight and activation traffic by half.

## 5. Empirical evaluations

### 5.1. End-to-end results

**Experimental setting.** We evaluate FlashSchNet on five fast-folding proteins following the benchmark suite of Charron et al. (2025): Chignolin (CLN, 10 residues), TRPcage (2JOF, 20 residues), Homeodomain (1ENH, 54 residues), Villin (1YRF, 35 residues), and Alpha3D (2A3D, 73 residues). All simulations use Langevin dynamics at 300 K with 64 parallel replicas and the step size of 4 fs on a single NVIDIA RTX PRO 6000 GPU. We compare against three baselines: CGSchNet (Charron et al., 2025) (the FP32 reference MLFF), the classical MARTINI force field (Marrink et al., 2007), and all-atom simulations. Structural fidelity is assessed via C$\alpha$ RMSD, fraction of native contacts

$Q$, and GDT-TS. Throughput is reported in timestep·mol/s (*i.e.*, simulation steps per second aggregated over all replicas). More details are included in Appendix B.

**Folding dynamics are preserved.** To verify that our optimizations preserve the physical fidelity of the underlying potential, we simulate Chignolin, TRPcage, and Villin for 16 ns each. Figure 4 shows the evolution of RMSD and $Q$ over simulation time. All trajectories exhibit multiple reversible folding transitions with the expected anti-correlation between RMSD and $Q$. Chignolin shows rapid nanosecond-scale transitions between folded ($Q > 0.8$) and unfolded ($Q < 0.4$) states; TRPcage exhibits dynamic fluctuations with $Q$ oscillating between 0.4 and 0.9; Villin displays longer residence times in metastable basins, reaching the native state ($Q > 0.85$) multiple times. These results confirm that `FlashSchNet` correctly samples the conformational landscape without introducing numerical artifacts.

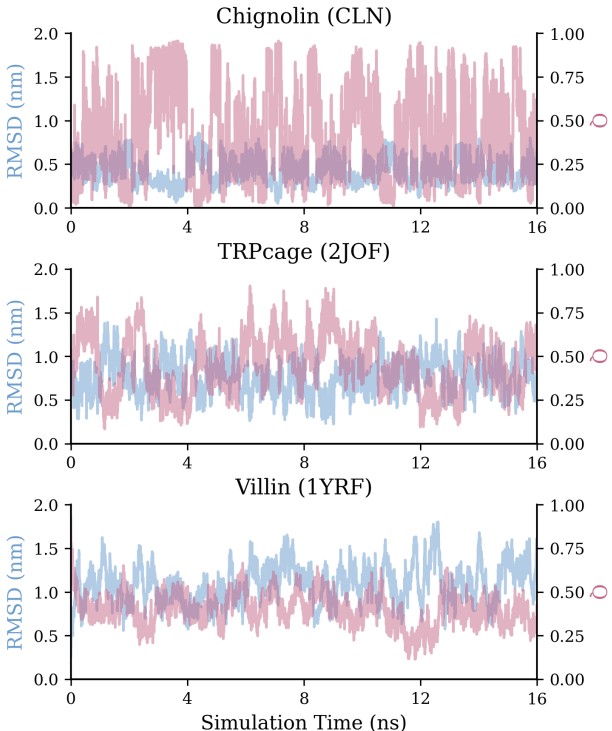

*Figure 4.* Trajectories of C$\alpha$ RMSD and fraction of native contacts ($Q$) for three fast-folding proteins simulated with `FlashSchNet`. The plots demonstrate multiple reversible folding/unfolding events with the expected anti-correlation between RMSD and $Q$. `FlashSchNet` successfully captures the distinct folding timescales of Chignolin (nanosecond transitions) compared to the longer residence times of TRPcage and Villin.

**Structural fidelity matches CGSchNet baseline.** Table 2 benchmarks structural accuracy using GDT-TS and the largest metastable $Q$. `FlashSchNet` maintains GDT-TS scores within 0.04 of the CGSchNet baseline across all

*Table 2.* Structural accuracy benchmark. `FlashSchNet` retains the high fidelity of the baseline CGSchNet and substantially outperforms the classical MARTINI model. All-Atom simulations serve as the experimental reference.

| Protein | Metric | MLFFs | | Classical | Reference |
|---|---|---|---|---|---|
| | | FlashSchNet | CGSchNet | MARTINI | All-Atom |
| Chignolin | GDT-TS | 0.90 | 0.90 | 0.66 | 1.00 |
| | Largest $Q$ | 0.94 | 0.96 | 0.83 | 0.95 |
| TRPcage | GDT-TS | 0.72 | 0.72 | 0.64 | 0.88 |
| | Largest $Q$ | 0.95 | 0.96 | 0.60 | 0.95 |
| Villin | GDT-TS | 0.74 | 0.78 | 0.46 | 0.88 |
| | Largest $Q$ | 0.96 | 0.94 | 0.56 | 0.93 |

proteins, while both MLFFs substantially outperform MARTINI in stabilizing near-native structures. After aligning the $Q$ computation with the CGSchNet C$\alpha$ contact protocol, `FlashSchNet` also matches the CGSchNet largest metastable $Q$ values and remains significantly higher than MARTINI ($Q \approx 0.56$–$0.83$). These findings confirm that `FlashSchNet` preserves the physical accuracy of the original CGSchNet model while improving simulation speed significantly.

*Table 3.* Computational efficiency benchmark. Evaluated proteins include Chignolin (CLN), TRPcage (2JOF), Homeodomain (1ENH), Villin (1YRF), and Alpha3D (2A3D). Performance metrics reported are speed (timestep·mol/s) and peak memory (GB). `FlashSchNet` demonstrates competitive throughput compared to classical benchmarks on a single RTX PRO 6000 GPU.

| Protein system | Metric | MLFF | | Classical | Reference |
|---|---|---|---|---|---|
| | | FlashSchNet | CGSchNet | MARTINI | All-Atom |
| Chignolin | Speed | **5222** | 3578 | 2580 | 1437 |
| | Peak Mem. | **3.7** | 22.7 | 35.0 | 36.3 |
| TRPcage | Speed | **4938** | 1729 | 2550 | 1419 |
| | Peak Mem. | **8.8** | 29.2 | 34.9 | 38.3 |
| Homeodomain | Speed | **3095** | 477 | 2250 | 1005 |
| | Peak Mem. | **18.0** | 92.5 | 34.9 | 37.4 |
| Villin | Speed | **3912** | 1056 | 2340 | 1275 |
| | Peak Mem. | **12.9** | 94.2 | 35.0 | 47.9 |
| Alpha3D | Speed | **2610** | 288 | 2160 | 861 |
| | Peak Mem. | **22.4** | 94.1 | 31.7 | 63.6 |

**Throughput reaches classical force field parity.** Table 3 summarizes throughput and memory usage. On the Homeodomain (1ENH) system, `FlashSchNet` achieves around **3000 timestep·mol/s** (*i.e.* 1000 ns/day), a **6.5× speedup** over the CGSchNet baseline (around 500 timestep·mol/s). This effectively closes the gap between MLFFs and classical potentials, as `FlashSchNet` reaches parity with MARTINI (around 2900 timestep·mol/s) while significantly outperforming all-atom simulations (around 1200 timestep·mol/s). This 1000 ns/day figure is aggregate throughput across 64 replicas; single-trajectory and replica-scaling details are reported in Appendix C. Moreover, `FlashSchNet` reduces peak memory from 92GB (CGSchNet) to **18GB** ($> 80\%$ **reduction**) by removing materialization of large intermediates. This potentially enables simulations of large systems on commodity hardware (*e.g.* a single RTX 5090).

## 5.2. Ablation results

**Robustness to dynamic graph topology.** A key challenge in GNN-MD is that neighbor graphs evolve throughout simulation, particularly during conformational transitions. As shown in Figure 6, the elongated 1ENH protein unfolds over 300k steps, causing the adjacency matrix to shift from near-diagonal to dense off-diagonal structure, with edge count increasing. Figure 5 reveals that CGSchNet throughput degrades substantially under these conditions, likely due to increased scatter contention when edges distribute across more destination nodes (Gong et al., 2025). In contrast, `FlashSchNet` maintains stable throughput via contention-free CSR segment reductions, which are agnostic to edge distribution patterns. This robustness is critical for practical MD workflows involving large conformational changes.

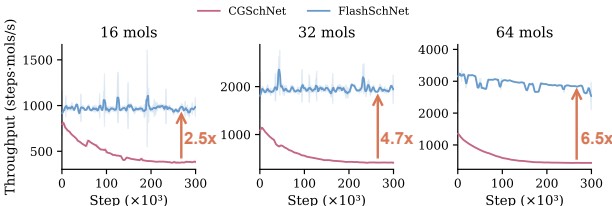

*Figure 5.* Step-wise throughput comparison on 1ENH protein during 300k-step elongated simulation across three batch sizes (*i.e.* 16, 32, 64 parallel replicas). `FlashSchNet` maintains consistent throughput along simulation despite evolving graph topology, while CGSchNet degrades as the neighbor graph becomes denser and less diagonal, as shown in Figure 6. The speedup gap widens with batch size, reaching $6.5\times$ at 64 replicas.

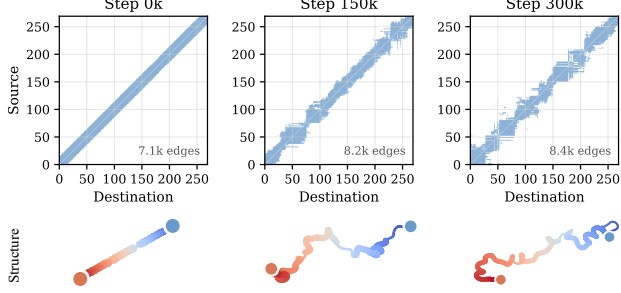

*Figure 6.* Evolution of graph topology and protein structure during 1ENH elongated simulation. Top: Adjacency matrices at steps 0, 150k, and 300k, showing increasing off-diagonal density as the protein unfolds (*e.g.* edges grow from 7.1k to 8.4k). Bottom: Corresponding 3D structures colored by residue index, illustrating the transition from compact folded state to extended conformations.

**Memory reduction enables better scalability.** As shown in Figure 7, we examine how throughput scales with the number of parallel replicas across four protein systems of varying size: Chignolin (CLN, 10 residues), Villin (1YRF, 35 residues), Homeodomain (1ENH, 54 residues), and Alpha3D (2A3D, 73 residues). CGSchNet exhausts GPU memory at small batch sizes across all systems (see insets), limiting its utility for enhanced sampling methods such as

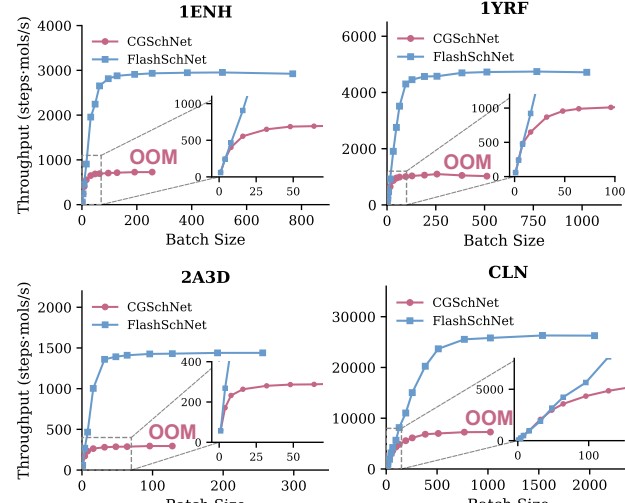

*Figure 7.* Throughput scaling with batch size across four protein systems on a single RTX PRO 6000 GPU. `FlashSchNet` consistently scales to $3$-$10\times$ larger batch sizes than CGSchNet before saturating, while CGSchNet encounters out-of-memory (OOM) at much smaller batch sizes (insets). The memory reduction from our IO-aware design is critical for enhanced sampling workflows requiring many parallel trajectories.

replica exchange that benefit from many concurrent trajectories. In contrast, `FlashSchNet` scales to $3$-$10\times$ larger batch sizes depending on system size, *i.e.* from $256$ replicas for the largest protein (Alpha3D) to $2048$ replicas for Chignolin. Throughput grows near-linearly before gradually saturating as compute resources become fully utilized. This scalability is particularly valuable for workflows requiring statistical convergence over many independent trajectories.

## 6. Conclusion

We present `FlashSchNet`, an IO-aware SchNet-style GNN molecular dynamics framework that addresses the memory-bound nature of learned potentials. By identifying HBM traffic as the key bottleneck rather than FLOPs, we developed four techniques that exploit inherent model structure to reduce data movement at the algorithmic level. *Flash radial basis* fuses distance computation and basis expansion into a single tiled pass. *Flash message passing* eliminates intermediate edge tensor materialization. *Flash aggregation* reformulates scatter-add via CSR segment reduce for contention-free accumulation. *Channel-wise 16-bit quantization* exploits low per-channel dynamic range to further improve throughput. Together, these techniques achieve **6.5×** **speedup** and **80% memory reduction** over the CGSchNet baseline, reaching 1000 ns/day aggregate throughput on coarse-grained protein containing 269 beads across 64 parallel replicas on a single RTX PRO 6000 GPU. To our knowledge, `FlashSchNet` is the first SchNet-style GNN-MD that is faster than classical coarse-grained force fields, *e.g.* MARTINI, in wall-clock efficiency while retaining the accuracy and transferability of learned potentials.

## Impact Statement

FlashSchNet improves the efficiency and memory footprint of SchNet-style GNN molecular dynamics through IO-aware fused kernels, contention-free aggregation, and lightweight quantization. These gains enable more concurrent multi-replica simulations on a fixed GPU budget, improving statistical efficiency and coverage of rare events. This can broaden access to accurate learned MD for academic and industrial users in computational chemistry, drug discovery, and materials science. By increasing utilization and reducing redundant memory movement, the techniques may also lower energy per simulated nanosecond, although net environmental impact depends on whether efficiency gains lead to more total simulation.

## Acknowledgements

Pingzhi Li, Hongxuan Li, and Tianlong Chen are partially supported by the NVIDIA Academic Grant Program and Amazon Research Award. Zirui Liu is partially supported by NSF CNS 2450525. Hongxuan Li and Xingcheng Lin are partially supported by NC State Startup Funding.

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

# A. Code

Our code is provided in https://github.com/unites-lab/flash-molecular-dynamics.

# B. Details of evaluation metrics and protocols

### B.1. Root Mean Square Deviation (RMSD)

The C$\alpha$ Root Mean Square Deviation (RMSD) quantifies the geometric deviation between a sampled conformation and a reference structure. Given $N$ C$\alpha$ atom positions $\{\mathbf{r}_i\}_{i=1}^N$ in the query structure and corresponding positions $\{\mathbf{r}_i^{\text{ref}}\}_{i=1}^N$ in the reference, the RMSD is defined as the minimum Euclidean distance achievable under rigid-body transformation:

$$\text{RMSD} = \min_{\mathbf{R}\in\text{SO}(3),\mathbf{t}\in\mathbb{R}^3} \sqrt{\frac{1}{N}\sum_{i=1}^N \|\mathbf{R}\mathbf{r}_i + \mathbf{t} - \mathbf{r}_i^{\text{ref}}\|^2} \tag{1}$$

where $\mathbf{R}$ denotes the rotation matrix and $\mathbf{t}$ the translation vector that optimally align the two structures. This superposition is typically computed via singular value decomposition (SVD). Lower RMSD values indicate higher structural fidelity to the native state.

### B.2. Fraction of Native Contacts ($Q$)

The fraction of native contacts ($Q$) serves as a reaction coordinate to quantify structural similarity based on pairwise residue distances, capturing topological fidelity rather than global superposition (Best et al., 2013). It is defined as:

$$Q = \frac{1}{N_c} \sum_{(i,j)\in\mathcal{C}} \frac{1}{1 + \exp\left[\beta(r_{ij} - \lambda r_{ij}^0)\right]} \tag{2}$$

where $\mathcal{C}$ denotes the set of native contact pairs, $N_c = |\mathcal{C}|$ is the total number of native contacts, $r_{ij}$ is the C$\alpha$ distance between residues $i$ and $j$ in the query structure, and $r_{ij}^0$ is the corresponding distance in the reference native structure.

**Native contact definition.** We define a native contact for any residue pair $(i, j)$ that satisfies two criteria in the reference all-atom structure: (1) a sequence separation $|i - j| \geq 3$, and (2) a heavy-atom distance less than 4.5 Å. The C$\alpha$ distances of these identified pairs form the reference set $\{r_{ij}^0\}$.

**Parameters.** Following (Charron et al., 2025), we set $\beta = 10\,\text{nm}^{-1}$ and $\lambda = 1.5$. The parameter $\beta$ modulates the steepness of the sigmoid function, while $\lambda$ accounts for thermal fluctuations around the native distance. These hyperparameters produce smooth free energy surfaces that clearly distinguish native-like states ($Q \approx 1$) from unfolded configurations ($Q < 0.5$).

**Largest metastable $Q$.** During simulation, the protein samples a probability distribution over $Q$. To evaluate the force field's ability to stabilize the native state, we compute the 1D probability density of $Q$, apply Savitzky-Golay smoothing, and identify the *rightmost* local maximum (i.e., the stable basin with the highest $Q$ value). This metric indicates the structural fidelity of the folded state populated by the model.

### B.3. GDT-TS Score

The Global Distance Test Total Score (GDT-TS) (Zemla, 2003) quantifies structural similarity by identifying the maximal subset of C$\alpha$ atoms that can be superimposed within a defined distance cutoff. For a specific cutoff $d$, let $P_d$ denote the percentage of C$\alpha$ atoms in the query structure falling within $d$ Å of their corresponding positions in the reference structure after optimal superposition. The GDT-TS is calculated as:

$$\text{GDT-TS} = \frac{P_1 + P_2 + P_4 + P_8}{4} \tag{3}$$

where $P_1$, $P_2$, $P_4$, and $P_8$ correspond to cutoffs of 1, 2, 4, and 8 Å, respectively. Unlike RMSD, GDT-TS is less sensitive to local high-variance regions (e.g., loops) and provides a robust metric for global topology. All GDT-TS calculations are performed using the TM-score program (Zhang & Skolnick, 2004).

**Evaluation protocol.** Following the protocol in (Charron et al., 2025), we construct a 2D free energy surface in RMSD vs. $Q$ space and apply $k$-means clustering with 100 centers. We identify the most native-like cluster (defined by the highest $Q$ and lowest RMSD) and randomly sample 10 representative structures from this basin. The reported GDT-TS is the average score of these samples against the experimental reference structure.

## C. Additional rebuttal results and clarifications

We summarize additional experiments and clarifications requested during review. Unless noted otherwise, throughput is reported in timestep·mol/s on a single NVIDIA RTX PRO 6000 GPU.

### C.1. NVE stability under W16A16

We evaluated Homeodomain (1ENH) in the NVE ensemble with 8 replicas over 1.6 ns trajectories (400k timesteps). W16A16 remains numerically comparable to the FP32 `FlashSchNet` baseline, indicating that reduced precision does not introduce measurable extra instability in this setting.

*Table 4.* NVE stability comparison on 1ENH.

| Approach | Throughput | $dE/dt$ (per ns) |
|---|---|---|
| `FlashSchNet` FP32 | 800.17 | $346 \pm 42$ |
| `FlashSchNet` W16A16 | 816.66 | $372 \pm 38$ |

### C.2. Larger-system scaling

We further evaluated larger protein systems up to 927 beads. The speedup remains high, and increases for several larger settings because scatter-add contention grows with edge count while CSR segment reduce remains contention-free.

*Table 5.* Throughput scaling on larger systems.

| Protein (replicas) | Beads | `FlashSchNet` | CGSchNet | Speedup |
|---|---|---|---|---|
| 1ENH (64) | 269 | 3095 | 477 | $6.5\times$ |
| 2A3D (64) | 360 | 2610 | 288 | $9.1\times$ |
| `1D3Z_ext` (16) | 508 | 831 | 94 | $8.9\times$ |
| `MCL1_ext` (16) | 927 | 394 | 45 | $8.7\times$ |

### C.3. CSR rebuild overhead

We rebuild the neighbor list every MD step in this profiling. The CSR layout construction uses bucket sort over edge endpoints, giving $O(E)$ construction cost with a small constant. Isolated profiling shows that CSR rebuild is below 1% of step time across the tested systems.

*Table 6.* Isolated CSR rebuild overhead.

| Config | Edges | Rebuild time per step | Step-time share |
|---|---|---|---|
| 1ENH (1 replica) | 30k | 0.15 ms | 0.52% |
| 1ENH (8 replicas) | 240k | 0.15 ms | 0.52% |
| 1ENH (64 replicas) | 1.9M | 0.28 ms | 0.97% |
| 2A3D (64 replicas) | 3.5M | 0.28 ms | 0.97% |
| MCL1 (16 replicas) | 2.0M | 0.27 ms | 0.93% |

### C.4. Replica scaling and ablations

Table 7 reports single-trajectory and multi-replica scaling on 1ENH. The single-trajectory case is already faster than CGSchNet, and the gap widens with batch size as `FlashSchNet`'s memory footprint grows slowly.

*Table 7.* 1ENH replica scaling. Memory is reported in GB.

| Batch size | FlashSchNet | CGSchNet | Speedup | Flash mem. | CG mem. |
|---|---|---|---|---|---|
| 1 | 124.98 | 89.31 | 1.40× | 2.93 | 0.46 |
| 4 | 502.72 | 311.22 | 1.62× | 2.94 | 6.46 |
| 16 | 982.35 | 425.48 | 2.31× | 3.53 | 93.37 |
| 32 | 2070.54 | 441.17 | 4.69× | 4.93 | 93.49 |
| 64 | 3095.35 | 476.97 | 6.48× | 11.40 | 93.49 |

Table 8 gives the cumulative contribution of each optimization on 1ENH with 64 replicas.

*Table 8.* Cumulative ablation on 1ENH with 64 replicas.

| Stage | Throughput | Speedup |
|---|---|---|
| Baseline | 477 | 1.0× |
| + Flash RBF | 799 | 1.7× |
| + Flash RBF + Flash MP | 1838 | 3.9× |
| + Flash RBF + Flash MP + CSR | 1967 | 4.1× |
| + Flash RBF + Flash MP + CSR + W16A16 | 2642 | 5.5× |
| Full FlashSchNet | 3095 | 6.5× |

## C.5. TorchMD-Net comparison and transferability

We also compare against TorchMD-Net 2.0 (Pelaez et al., 2024) on 1ENH. TorchMD-Net 2.0 optimizes the broader simulation stack, while FlashSchNet targets the SchNet-style GNN kernels and therefore gives complementary acceleration in this coarse-grained setting.

*Table 9.* Throughput comparison with TorchMD-Net 2.0 on 1ENH.

| Batch size | FlashSchNet | CGSchNet | TorchMD-Net 2.0 |
|---|---|---|---|
| 16 | 982 | 425 | 431 |
| 32 | 2071 | 441 | 438 |
| 64 | 3095 | 477 | 446 |

The techniques have different levels of architectural specificity. CSR segment reduce applies to GNNs with sum aggregation, including architectures such as MACE (Batatia et al., 2022), NequIP (Batzner et al., 2022), and DimeNet (Gasteiger et al., 2020). Flash radial basis applies to models that compute radial bases from pairwise distances. W16A16 applies to MLP submodules after validating per-architecture dynamic range. The fused message-passing kernel is the most SchNet-specific component; extending the same IO-aware principle to equivariant tensor-product pipelines is complementary to FlashTP (Lee et al., 2025) and cuEquivariance (NVIDIA, 2024). Universal models such as UMA (Wood et al., 2025) further highlight the demand for fast learned-potential inference, while evaluation on larger biomolecular assemblies, all-atom and materials workloads, other GPU generations, and portable backends remains future work.

