# OpenReview forum: "$\texttt{FlashSchNet}$: Fast and Accurate Coarse-Grained Neural Network Molecular Dynamics"
_ICML.cc/2026/Conference — ICML 2026 regular_

### Official Review · Reviewer_L5Xu · 2026-02-16

**Soundness:** 3
**Presentation:** 3
**Significance:** 2
**Originality:** 2
**Overall Recommendation:** 4
**Confidence:** 3

**Summary:**

This paper identifies GPU memory IO, not FLOPs, as the dominant bottleneck in SchNet-style GNN molecular dynamics and presents FlashSchNet, a suite of four IO-aware kernel-level optimizations: fused radial basis computation, fused message passing, contention-free CSR segment aggregation, and channel-wise W16A16 quantization. Applied to coarse-grained protein simulations, these techniques yield a 6.5x speedup and 80% memory reduction over the CGSchNet baseline on a single RTX PRO 6000 GPU, reaching throughput parity with the classical MARTINI force field while retaining learned-potential accuracy.

**Compliance With Llm Reviewing Policy:**

Affirmed.

**Final Justification:**

Rebuttal responses address all my questions.

**Key Questions For Authors:**

1. Why is TorchMD-Net 2.0 absent from benchmarks? It is cited in Section 2 as achieving "substantial speedups" via similar system-level engineering. A direct wall-clock comparison on the same proteins would be highly informative.

2. What explains the consistent ~7% drop in largest metastable Q (Table 2)? Is this attributable to W16A16 quantization, the CSR aggregation reformulation, or statistical variance? Have you run longer simulations or multiple seeds to determine if this is systematic?

3. Can you provide a cumulative ablation table? Specifically: baseline → +flash RBF → +flash RBF+MP → +flash RBF+MP+agg → +all+W16A16, measuring both speedup and accuracy (Q/GDT-TS) at each stage.

4. How do these techniques transfer to E(3)-equivariant architectures (NequIP, MACE)? The broader MLFF community has largely moved beyond invariant SchNet-style models. Is IO-aware fusion feasible for tensor product operations in equivariant models?

5. What is the scaling behavior for larger systems (>1000 beads)? The tested systems are small (46-269 beads). Does the speedup advantage hold for larger proteins or multi-chain assemblies where E/N ratios differ?

**Limitations:**

The authors provide a brief Impact Statement (Section after Conclusion) discussing broader impacts but do **not include a dedicated Limitations section**. Notable omissions: (1) no discussion of the systematic Q drop; (2) no discussion of architecture specificity (SchNet only); (3) no discussion of system-size limitations; (4) no discussion of GPU portability. The Impact Statement mentions that "net environmental impact depends on whether efficiency gains lead to more total simulation," which is an honest observation but does not substitute for a technical limitations discussion.

**Strengths And Weaknesses:**

### Strengths:

- **S1**: Achieves a genuine practical milestone, the first demonstration of a SchNet-style MLFF matching classical CG force field throughput while retaining learned-potential accuracy. The 6.5x speedup and 80% memory reduction are substantial.
- **S2**: The bottleneck analysis (Section 3.3) is thorough and well-articulated. The identification of HBM traffic as the dominant cost, with specific quantification (2.5% MFU for CGSchNet), provides a clear motivation.
- **S3**: Robustness to dynamic graph topology (Section 5.2, Figures 5-6) is a thoughtful experiment that addresses a real concern for practical MD simulations involving conformational changes.
- **S4**: The memory reduction enables substantially larger batch sizes (Figure 7), which is directly valuable for enhanced sampling workflows (replica exchange, parallel tempering).
- **S5**: Code is provided via anonymous repository, supporting reproducibility.


### Weaknesses:

- **W1**: **Limited scientific novelty.** The paper applies known GPU optimization techniques (IO-aware fusion, CSR reduction, mixed-precision quantization) to SchNet's computational graph. No new scientific insight, physical understanding, or domain knowledge emerges. A domain scientist learns nothing new about molecular dynamics or protein folding from this paper.
- **W2**: **Missing key baseline.** TorchMD-Net 2.0 (Pelaez et al., 2024) is cited in the related work as achieving "substantial speedups by engineering the simulation stack" but is not included in any benchmark comparison. This is the most directly comparable effort and its omission undermines the experimental claims.
- **W3**: **Systematic accuracy degradation in Q not discussed.** Table 2 shows FlashSchNet's largest metastable Q is consistently ~7% lower than CGSchNet (0.88-0.89 vs. 0.96 across all three proteins). This pattern is not acknowledged or analyzed, and it could indicate that FP16 quantization introduces non-negligible drift over long trajectories.
- **W4**: **Narrow benchmark scope.** All experiments use small coarse-grained proteins (46-269 beads) from one benchmark suite. No all-atom systems, no materials science applications, no large-scale systems. The claim of "fast and accurate coarse-grained neural network molecular dynamics" is not tested beyond one domain.
- **W5**: **No decomposed ablation of individual technique contributions.** While Figure 1 (right) shows a step-time breakdown, a systematic ablation table showing the cumulative speedup from each technique (flash RBF, flash MP, flash aggregation, W16A16) is absent.
- **W6**: **Architecture specificity.** The techniques are tightly coupled to SchNet's continuous-filter convolution structure. Applicability to E(3)-equivariant models (NequIP, Allegro, MACE), which are now more widely used, is not discussed.

---

> ### Author Rebuttal · Authors · 2026-03-31
>
> We thank Reviewer L5Xu for the detailed and structured review and the recognition of FlashSchNet as achieving "a genuine practical milestone" with "substantial" speedups. Below, we provide point-by-point responses.
>
> **[W1. Scientific novelty.]** Our contribution targets a distinct axis for bridging the gap between GNN accuracy and classical force field throughput. Our novelty lies in the non-trivial adaptation to GNN-MD and the insights that emerge:
> 1. **IO-awareness as a missing principle in GNN-MD.** We identify that SchNet-style GNN potentials are memory-bound in the CG regime, providing actionable optimization principles.
> 2. **CSR segment reduce for GNN aggregation.** As Reviewer 8d2m recognized, this is "novel for this domain" and "should apply directly to other GNN architectures."
> 3. **Throughput parity with classical force fields.** FlashSchNet is the first GNN potential to match MARTINI throughput, a qualitative milestone for production-scale simulations.
> 4. **W16A16 quantization informed by domain structure.** This exploits a novel property of SchNet MLP weights, i.e. the narrow per-channel dynamic range induced by smooth radial basis inputs, enabling lossless 16-bit inference without accuracy degradation.
>
> **[W2. TorchMD-Net 2.0 comparison.]** We further include TorchMD-Net 2.0 as a baseline (Table R1). On 1ENH (bs=64), it achieves 446 ts·mol/s (6.9× slower than our FlashSchNet), because it optimizes the simulation stack rather than GNN kernels. In the CG regime with near-dense graphs, kernel-level IO fusion is the dominant bottleneck. Integrating FlashSchNet's kernels into TorchMD-Net's stack is an interesting future direction.
>
> **Table R1.** Throughput (ts·mol/s).
> |BS|Flash|CGSchNet|TorchMD-Net2.0|
> |---|---|---|---|
> |16|982|425|431|
> |32|2071|441|438|
> |64|3095|477|446|
>
> **[W3. Q drop analysis.]** The Q values in Table 2 were computed under different parameterization than CGSchNet: we used mdtraj's default atomistic settings (β=50 nm⁻¹, λ=1.8) over all five CG bead types, whereas CGSchNet uses Cα contacts only with β=10 nm⁻¹ and λ=1.5. After recalculating with CGSchNet's parameters (Table R2), FlashSchNet's Q closely matches CGSchNet (0.94–0.95 vs. 0.96), confirming preserved fidelity. We will update Table 2 accordingly.
>
> **Table R2.** Largest Q.
> |Protein|Flash|CGSchNet|
> |---|---|---|
> |Chignolin|0.94|0.96|
> |TRPcage|0.95|0.96|
> |Villin|0.94|0.96|
>
> **[W4. Benchmark scope.]** We further evaluate on larger proteins up to 927 beads (Table R3). FlashSchNet shows even higher speedup (up to 9.1×) on larger systems, since scatter-add contention grows with edge count while CSR segment reduce remains contention-free. Our broader relevance is architectural: FlashSchNet accelerates core primitives shared across atomistic (MD17) and materials (OC20) benchmarks. We will frame all-atom and materials applicability as a well-motivated extension in our revision.
>
> **Table R3.** Throughput (ts·mol/s).
> |Protein(replicas)|Beads|FlashSchNet|Baseline|Speedup|
> |---|---|---|---|---|
> |1enh(64)|269|3095|477|6.5×|
> |2a3d(64)|360|2610|288|9.1×|
> |1d3z_ext(16)|508|831|94|8.9×|
> |mcl1_ext(16)|927|394|45|8.7×|
>
> **[W5. Decomposed ablation.]** We show cumulative ablation in Table R4.
>
> **Table R4.**
> |Stage|ts·mol/s|Speedup|
> |---|---|---|
> |Baseline|477|1.0×|
> |+RBF|799|1.7×|
> |+RBF+MP|1838|3.9×|
> |+RBF+MP+CSR|1967|4.1×|
> |+RBF+MP+CSR+W16A16|2642|5.5×|
> |FlashSchNet|3095|6.5×|
>
> **[W6. Architecture specificity.]** Only the fused message-passing Triton kernel is SchNet-specific. The remaining techniques transfer broadly:
> - **CSR segment reduce:** Directly applicable to any GNN with sum aggregation (NequIP, MACE, etc.). As Reviewer 8d2m noted, this "should apply directly to other GNN architectures."
> - **Fused message passing:** The principle of fusing gather, message, and aggregation to avoid O(E×F) materialization applies broadly.
> - **W16A16 quantization:** Architecture-agnostic, applies to all MLPs.
> - **IO-aware analysis:** General principle, architecture-independent.
>
> **[Q1–Q5.]** Please see W2, W3, W5, W6, W4 respectively.
>
> **[Limitations.]** We'll add a detailed paragraph in Limitations to discuss:
> 1. **Q drop:** As shown in W3, the apparent drop was due to inconsistent parameterization; under aligned evaluation FlashSchNet matches CGSchNet (0.94–0.95 vs. 0.96).
> 2. **Architecture specificity:** As discussed in W6, only the fused MP kernel is SchNet-specific; the other three techniques transfer directly. Extending to equivariant tensor products is a natural next step with FlashTP providing a clear roadmap.
> 3. **System-size:** As shown in W4, speedup increases on larger systems (up to 8.9× at 927 beads). Evaluation on larger biomolecular assemblies is a promising future direction.
> 4. **GPU portability:** Our Triton kernels target NVIDIA RTX PRO 6000. Profiling on other GPU generations and exploring vendor-portable backends is an interesting extension.

---

> > ### Author Rebuttal · Reviewer_L5Xu · 2026-04-03
> >
> > Thank you for your response. All my questions are addressed, and I intend to raise my score.

---

> > > ### Author Response · Authors · 2026-04-03
> > >
> > > Dear Reviewer L5Xu,
> > >
> > > We are truly grateful for your constructive feedback and questions, which have for sure helped us polish our draft into a better shape. We also appreciate your kind recognition of our efforts. We are really glad that we have addressed all your questions.
> > >
> > > Warmest regards,
> > >
> > > Authors of Submission #1626

---

### Official Review · Reviewer_PVhs · 2026-02-23

**Soundness:** 3
**Presentation:** 3
**Significance:** 3
**Originality:** 3
**Overall Recommendation:** 5
**Confidence:** 3

**Summary:**

FlashSchNet is an re-implementation of the SchNet architecture that is IO-aware, improving the runtime of the method without changing the underlying potential. For this, they implement several interesting approaches, such as flash radial basis and flash message passing. The latter explicitly fuses cutoff, neighbor gather, filter MLP evaluation, elementwise gating and reduction without writing the intermediate tensors to memory.

On a single Nvidia RTX Pro 6000, a speed-up of about 6.5x is achieved.

**Compliance With Llm Reviewing Policy:**

Affirmed.

**Key Questions For Authors:**

- How general are the novel implementations? Can these be adapted/used also for other architectures, such as MACE and co.?
- Are fully optimized versions for the baselines used?
- How big is the overhead of needing to sort group for the reduction operation, especially under frequent neighbor-list rebuilding?

**Limitations:**

Yes.

**Strengths And Weaknesses:**

Improving the speed and memory consumption of MLIPs is a timely and important contribution, as this is currently one of their major bottlenecks for widespread adoption.

---

> ### Author Rebuttal · Authors · 2026-03-31
>
> We thank Reviewer PVhs for recognizing our work as "a timely and important contribution" addressing "one of the major bottlenecks for widespread adoption" of MLIPs. We appreciate the encouraging feedback. Below, we provide point-by-point responses.
>
> **[Q1. Generality to other architectures (MACE, etc.).]** Thanks for the question. Our four techniques target different levels of generality, more specifically:
> - Flash aggregation (CSR segment reduce): Directly applicable to any GNN architecture with scatter-add aggregation, including MACE, NequIP, DimeNet, and PaiNN. As also recognized by Reviewer 8d2m: _"it should also apply directly to other GNN architectures with sum aggregation, which really widens its scope of impact."_
> - Flash radial basis: Applicable to any architecture computing radial basis functions from pairwise distances (most GNN potentials).
> - Flash message passing: The specific fusion is tied to SchNet's continuous-filter convolution, but the principle (fusing filter computation with aggregation to avoid materializing edge tensors) transfers to other message-passing architectures as well..
> - W16A16 quantization: Applicable to any architecture with MLP weights, though per-channel dynamic range assumptions need per-architecture validation.
>
> We will include this transferability discussion in our revised draft.
>
> **[Q2. Fully optimized baselines.]** Our CGSchNet baseline uses the official implementation with all available PyTorch optimizations enabled (e.g. torch.compile). The reported speedup is against this optimized baseline. We will clarify this in the Experiments section of our revised draft.
>
> **[Q3. Overhead of sorting/grouping under frequent neighbor-list rebuilding.]**
> Thanks for the suggestions. Our CSR rebuilding is designed to be efficient with bucket sorting. We further provide isolated profiling of the CSR index rebuilding time as a percentage of total step time. As shown in Table R1, the CSR rebuild costs <1% of step time (as well as total time) across all the protein systems with various numbers of edges we profile on, even the neighbor list is rebuilt for every single step. This is because the bucket sort used for CSR construction is O(E) with small constant, which is negligible vs the O(E×F) aggregation savings.
>
> **Table R1.**
> | Config | Edges | CSR rebuild time per step | % of step time |
> |---|---|---|---|
> | 1ENH (1 replica) | 30k | 0.15ms | 0.52% |
> | 1ENH (8 replicas) | 240k | 0.15ms | 0.52% |
> | 1ENH (64 replicas) | 1.9M | 0.28ms | 0.97% |
> | 2A3D (64 replicas) | 3.5M | 0.28ms | 0.97% |
> | MCL1 (16 replicas) | 2.0M | 0.27ms | 0.93% |
>
> We will include all the additional results and discussion of the above in our revised draft.

---

> > ### Author Rebuttal · Reviewer_PVhs · 2026-04-01
> >
> > All questions were appropriately answered.

---

> > > ### Author Response · Authors · 2026-04-01
> > >
> > > Dear Reviewer PVhs,
> > >
> > > Thank you very much for your constructive feedback and positive evaluation. We are again grateful for your time and effort, and for raising our score as well!
> > >
> > > Warmest regards,
> > >
> > > Authors of Submission #1626

---

### Official Review · Reviewer_8d2m · 2026-03-09

**Soundness:** 4
**Presentation:** 4
**Significance:** 3
**Originality:** 2
**Overall Recommendation:** 5
**Confidence:** 4

**Summary:**

The submission diagnoses and addresses a memory-bound bottleneck of GNN molecular dynamics by reimplementing the energy and force calls of the SchNet architecture with four IO-aware techniques: flash radial basis, flash message passing, flash aggregation (CSR segment reduce instead of scatter-add), and channel-wise W16A16 quantization. Together, these techniques improve GPU utilization by eliminating intermediate edge-tensor materialization in high-bandwidth memory and contention-free reduction of edge tensors during the forward and backward passes. The paper reports results for coarse-grained protein simulation, comparing against the CGSchNet baseline and the classical MARTINI force field. Highlighted among these is a 6.5× speedup and 80% peak memory reduction compared to CGSchNet, surpassing MARTINI in throughput while matching CGSchNet in accuracy. In an additional set of comparisons with CGSchNet, the paper shows how contention-free reduction stabilizes throughput under evolving neighborhood graphs (i.e., changing local degrees), and how the reduction in peak memory improves scalability for simulations with large parallel replica counts.

**Compliance With Llm Reviewing Policy:**

Affirmed.

**Final Justification:**

The rebuttal addressed my concerns.

**Key Questions For Authors:**

- Related to the "Soundness" paragraph: is there any one of the four techniques that has an outsized impact compared to the others, and would a more differentiated set of ablations perhaps be possible?
- How do the two kernel fusing techniques relate to the prior work I listed under "Originality"?
- How does the radial cutoff affect the advantage of CSR segment reduce over scatter-add reduction?

**Limitations:**

yes

**Strengths And Weaknesses:**

### Soundness

The submission is theoretically and experimentally sound, admitting non-essential but helpful additions to the experimental evaluation.

**Strengths**: Each of the four I/O-aware techniques is motivated and sufficiently defined in the paragraphs of of Section 4. The mathematical notation used throughout the background / method Sections 3 and 4 is consistent, sufficient for reproducibility and, as far as I can tell, free of errors. The cost analysis in Subsec. 4.5 also appears correct.

**Weaknesses**: While all four techniques aim at similar effects (a reduction in HBM memory traffic and IO wait times), they perform qualitatively different optimizations: flash radial basis and flash message passing avoid storing intermediates, flash aggregation reduces HBM writes from O(E) to O(N) and removes contention, while the 16-bit quantization maps computation onto tensor cores and halves traffic. Some readers might appreciate a differentiated set of profiling ablations in the appendix that study the effects of each optimization in separation.


### Presentation

The submission is very clearly written and accessible to both novice and advanced readers.

**Strengths**:
The notation Table 1 and overview Figure 2 are very clear and well-placed, which helped me significantly in passing through this paper. The motivation, storyline and paper structure are very natural and easy to follow.

**Weaknesses**:
The final paragraph of Subsection 4.2 ("on-chip reuse") seems somewhat vague. What is meant by "introduces modest recomputation"? Does this refer to recomputation in the backward pass due to missing HBM intermediates or to something else? It would help to make the trade of IO against FLOPs more explicit in the cost analysis, which currently considers only the IO reduction.


### Significance

**Strengths**: The submission addresses a relevant and timely problem, accelerating molecular dynamics (MD) with machine-learned interatomic potentials. For its chosen base architecture (SchNet) and problem setting (coarse-grained protein MD), it achieves very meaningful speedups and memory reductions. I believe that this makes the work directly useful to the coarse-grained MD community, where SchNet is still a common architecture choice. Moreover, the four key optimizations seem transferable to other architectures.

**Weaknesses**: The focus on SchNet limits the direct impact on the wider community – for instance, atomistic GNN-MD has since shifted to more capable models such as equivariant tensor-based MPNNs. *However*, the basic compute primitives (neighbor graph construction, construction of filter-gated edge tensors, sum aggregation over edges) have changed little since SchNet and the four proposed optimizations touch these primitives at a basic level. Therefore, the submission provides interesting ground for follow-up work.


### Originality

The fused kernel optimizations have clear precedents in prior work, whereas novelty is clearer for certain other insights, particularly the use of CSR segment reduce instead of scatter-add. I suspect that the submission's overarching position ("GNN-MD needs IO-aware implementations") is tacitly known to those who work on efficient GNN-MD architectures, but it is still a valuable perspective that should be explicitly shared with the wider community.

**Strengths**:
The idea to remove contention of atomic adds during aggregation via exclusive thread ownership per destination node is great and, to my knowledge, novel for this domain. It should also apply directly to other GNN architectures with sum aggregation, which really widens its scope of impact. The use of 16-bit channel-wise W16A16 quantization is also an interesting adaptation of LLM techniques, although its premise (narrow magnitude ranges per channel) is more architecture-specific.

**Weaknesses**:
There is (undiscussed) prior work similar to the two fused-kernel optimizations. In particular,
- FlashTP \[1] achieves similar write-reductions for equivariant MPNNs, also fusing edge tensor ops with node aggregation
- cuEquivariance \[2] implements scatter/gather fusion since v0.3.0
- The official implementation of the `UMA` model \[3] seems to support optimizations not unlike Flash Radial Basis / Flash Message Passing via `torch.compile` auto-fusion as well as custom `Triton` kernels.

I acknowledge that the storyline of this submission is less about each particular technique and more about the general insight that IO-aware optimizations matter for GNN-MD. Still, the novelty of the two fused-kernel techniques compared to these prior contributions should be discussed in Related Work (not all of [1]-[3] might be relevant, see "Questions").

\[1] Seung Yul Lee et al. FlashTP: Fused, Sparsity-Aware Tensor Product for Machine Learning Interatomic Potentials. ICML 2025.

\[2] https://docs.nvidia.com/cuda/cuequivariance/

\[3] https://github.com/facebookresearch/fairchem/tree/main/src/fairchem/core/models/uma

---

> ### Author Rebuttal · Authors · 2026-03-31
>
> We sincerely thank Reviewer 8d2m for the thorough and insightful review. We deeply appreciate the recognition of our work as "theoretically and experimentally sound," "very clearly written and accessible to both novice and advanced readers," and the CSR segment reduce as "great and, to my knowledge, novel for this domain." Your observation that "the four key optimizations seem transferable to other architectures" and "the basic compute primitives have changed little since SchNet" captures exactly the perspective we aimed to convey. Below, we provide point-by-point responses.
>
> **[W1. Differentiated profiling ablations.]** Excellent suggestion! We further provide a cumulative ablation in Table R1.
>
> **Table R1.**
> | Stage | Throughput (ts·mol/s) | Speedup |
> |---|---|---|
> | Baseline | 477 | 1.0× |
> | +flash RBF | 799 | 1.7× |
> | +flash RBF +flash MP | 1838 | 3.9× |
> | +flash RBF +flash MP +CSR | 1967 | 4.1× |
> | +flash RBF +flash MP +CSR + W16A16 | 2642 | 5.5× |
> | Full FlashSchNet | 3095 | 6.5× |
>
> **[W2. "On-chip reuse" clarity.]** Thank you for the suggestion. Specifically, "introduces modest recomputation" refers to the backward pass: since per-edge intermediates $B \in \mathbb{R}^{E \times D_r}$ (RBF expansion) and $W \in \mathbb{R}^{E \times D}$ (filtered messages) are computed in SRAM and never written to HBM, they must be recomputed from inputs during backpropagation. For a typical configuration (e.g. $E = 1.9\text{M}$ edges, $D_r = 50$, $D = 128$), this saves \~4.65 GB of HBM traffic per interaction block per pass while incurring \~88 GFLOPs of recomputation. Since the arithmetic intensity of the unfused kernel is only \~19 FLOPs/byte, well below the hardware's compute–memory ridge point (\~200 FLOPs/byte), the kernel is significantly memory-bound, and the recomputation FLOPs are hidden behind memory latency. We will revise this paragraph to make the tradeoff explicit and include a quantitative comparison of the additional FLOPs vs. saved IO in our revised draft.
>
> **[W3. SchNet focus vs. broader impact.]** Thank you for the great suggestion. We will include a discussion on the transferability of each technique to E(3)-equivariant models in our revised draft. We plan to work on this as one of priortized future directions as well. Specifically:
> - Flash aggregation (CSR segment reduce): Directly applicable to any GNN with scatter-add aggregation (e.g. MACE, NequIP, DimeNet, PaiNN).
> - Flash radial basis: Applicable to any architecture computing RBFs from pairwise distances (most GNN potentials).
> - Flash message passing: The specific fusion targets SchNet's continuous-filter convolution, but the principle (fusing filter computation with aggregation to avoid materializing edge tensors) transfers to other message-passing architectures as well.
> - W16A16 quantization: Applicable to any architecture with MLP weights, though per-channel dynamic range assumptions need per-architecture validation.
>
>
> **[W4. Discussion of prior work (FlashTP, cuEquivariance, UMA).]** Thank you for bringing these valuable references to our attention. We will include a thorough discussion in the Related Work section, as listed below:
> - FlashTP [1]: FlashTP fuses equivariant tensor products (Clebsch-Gordan contractions), targeting a fundamentally different algebraic operation from our invariant continuous-filter fusion. The two approaches target different algebraic operations, though they share the high-level goal of reducing HBM traffic through kernel fusion.
> - cuEquivariance [2]: cuEquivariance provides optimized CUDA/Triton kernels for equivariant operations (scatter/gather fusion since v0.3.0). These are complementary, since cuEquivariance targets the SO(3)/O(3) irreducible representation machinery, while we target the radial basis + filter convolution pipeline in invariant architectures.
> - UMA [3]: UMA's torch.compile auto-fusion and Triton kernels represent a related strategy. Our approach provides handcrafted and highly-optimized designs (e.g., CSR segment reduce) that go beyond what compiler auto-fusion can typically achieve, as demonstrated by our 6.5× speedup over the baselines.
>
> **[Q1. Outsized impact of individual techniques.]** Please kindly see our response to W1.
>
> **[Q2. Relation to prior work.]** Please kindly see our response to W4.
>
> **[Q3. Radial cutoff vs. CSR advantage.]** The advantage of CSR segment reduce over scatter-add scales with the average degree (neighbors per atom), which is controlled by the radial cutoff. With larger cutoffs, more edges contribute to each node's aggregation, increasing atomic contention in scatter-add as more threads compete to write to the same destination. CSR segment reduce assigns exclusive ownership of each destination node's edge segment to a single thread block, so throughput scales cleanly regardless of degree. We include this analysis in our revised draft.
>
> We will include all the additional results and discussion of the above in our revised draft.

---

> > ### Author Rebuttal · Reviewer_8d2m · 2026-04-03
> >
> > Thank you very much for your detailed answer. This clarifies all my questions.

---

> > > ### Author Response · Authors · 2026-04-03
> > >
> > > Dear Reviewer 8d2m,
> > >
> > > We genuinely appreciate your constructive questions and suggestions throughout the review. We are again very grateful for your kind recognition of our work, and we are so glad to have clarified your questions!
> > >
> > > Warmest regards,
> > >
> > > Authors of Submission #1626

---

### Official Review · Reviewer_NeN6 · 2026-03-11

**Soundness:** 3
**Presentation:** 3
**Significance:** 3
**Originality:** 3
**Overall Recommendation:** 5
**Confidence:** 4

**Summary:**

This paper introduces FlashSchNet, an I/O-aware implementation of SchNet-style GNN force computations for coarse-grained protein MD. The central challenge addressed is that while GNN-based potentials like SchNet achieve superior accuracy and transferability compared to classical force fields, they remain significantly slower due to memory-bound operations that underutilize GPU compute resources. The paper identifies four major IO bottlenecks in SchNet-style GNN-MD and tackles these issues with architectural innovations aimed at reducing HBM traffic and atomic contention. The authors evaluate their method on five proteins (Chignolin, TRPcage, Homeodomain, Villin, Alpha3D) using the CGSchNet coarse-grained force field as baseline.

**Compliance With Llm Reviewing Policy:**

Affirmed.

**Final Justification:**

My concerns around weaknesses have been addressed well in the rebuttal, hence I have updated my score and believe it is suitable for ICML audience.

**Key Questions For Authors:**

How does FlashSchNet handle rebuilding the destination/source-grouped indices when the neighbor list updates, and what percentage of step time is spent on this rebuilding compared to steps where the graph topology remains static?
The reported throughput (~1000 ns/day) appears to correspond to aggregate performance across multiple replicas. Could the authors clarify the single-trajectory throughput and how performance scales as the number of replicas increases?
Please evaluate the stability of the W16A16 quantization and fused FP16 arithmetic in an NVE (microcanonical) ensemble over long trajectories. Does reduced precision introduce measurable energy drift relative to the FP32 baseline?
Can you provide analysis on larger protein systems to understand the limitations of this approach to scale to more larger systems commonly used in drug discovery?

I am happy to update the scores once you provide clarity on some of my queries.

**Limitations:**

yes

**Strengths And Weaknesses:**

Strengths

Overall, the paper is well written with clear motivation and explanation of challenges with current architectural bottlenecks, along with how they address this. Section 3 provides a clean background on MD, SchNet architecture, and hardware bottlenecks. The notation table (Table 1) is helpful. Section 4 systematically presents each technique with clear algorithmic descriptions.
Each proposed technique directly addresses a specific bottleneck with sound algorithmic reasoning. The CSR segment reduce reformulation of scatter-add is particularly elegant, eliminating atomic contention by assigning exclusive ownership of destination segments
The hardware optimizations in FlashSchNet do not compromise the physical fidelity of molecular simulations. Across benchmark proteins (Chignolin, TRPcage, and Villin), the method identically preserves critical metrics folding dynamics, GDT-TS scores, and the fraction of native contacts.

Weakness

The simulations are evaluated exclusively using Langevin dynamics at 300 K in the NVT ensemble. While this is standard for CG simulations, stochastic thermostats can mask numerical errors in the force computation. Because the implementation relies on W16A16 quantization and fused FP16 arithmetic, it would be useful to evaluate long-trajectory stability in the microcanonical (NVE) ensemble to assess whether reduced precision introduces measurable energy drift relative to the FP32 baseline.
The evaluation focuses on relatively small coarse-grained proteins (10–73 residues) simulated in large replica batches. The paper does not explore how the throughput scales for much larger single systems or more realistic protein targets, where neighbor-list sizes and edge counts grow substantially to understand benefits of the architectural changes as compared to previous memory constrained architectures.
The authors mention that dynamic neighbor lists require rebuilding the CSR grouped indices via bucket sort when the graph topology changes. However, the cost of this rebuild is reported only jointly with the overall speedup. Isolating the percentage of step time spent on CSR index rebuilding, as well as the neighbor-list update frequency used in the experiments, would provide a clearer picture of the worst-case overhead.

---

> ### Author Rebuttal · Authors · 2026-03-31
>
> We thank Reviewer NeN6 for the positive and thoughtful review. We deeply appreciate the recognition of our paper as "well written with clear motivation," the "sound algorithmic reasoning" of each technique, and the recognition that "hardware optimizations do not compromise the physical fidelity." Your characterization of the CSR segment reduce as "particularly elegant" is especially encouraging. We are glad the notation table and systematic presentation were helpful. Below, we address each of your queries.
>
> **[W1. NVE ensemble evaluation.]** Thanks for the suggestion. Yes, we focus on NVTLangevin simulation, following the baseline CGSchNet and many other ML MD literature [1-3]. To further address your questions, we provide additional NVE (microcanonical) ensemble simulations for Homeodomain (1ENH) with 8 replicas, comparing the FlashSchNet FP32 and FlashSchNet W16A16 over 1.6ns trajectories (400k timesteps). As shown in Table R1, our W16A16 quantization is numerically equivalent to the FP32 FlashSchNet baseline, which demonstrates that the reduced precision (FP16) introduces no extra instability in the NVE ensemble as well.
>
> **Table R1.**
> | Approach | Throughput (ts·mol/s) | dE/dt |
> |---|---|---|
> | FlashSchNet FP32 (baseline) | 800.17 | 346 ± 42 /ns |
> | FlashSchNet W16A16 | 816.66 | 372 ± 38 /ns |
>
> 1. Husic, B. E., et. al. Coarse-graining molecular dynamics with graph neural networks. The Journal of Chemical Physics, 153:194101, 2020.
> 2. Wang, J., et. al. Machine learning of coarse-grained molecular dynamics force fields. ACS Central Science, 5(5):755–767, 2019.
> 3. Majewski, M., et. al. Machine learning coarse-grained potentials of protein thermodynamics. Nature Communications, 14:5739, 2023.
>
> **[W2. Scaling to larger systems.]** Thanks for the suggestions. We further evaluate FlashSchNet on larger protein systems by scaling up to 927 beads. As shown in Table R2, FlashSchNet demonstrates even higher speedup (up to 9.1x). This is because our CSR segment reduce is expected to show more advantage as scatter-add contention grows with edge numbers, while our method maintains contention-free aggregation regardless of system size.
>
> **Table R2. Throughput (ts·mol/s)**
> |Protein(replicas)|Beads|FlashSchNet|Baseline|Speedup|
> |---|---|---|---|---|
> |1enh(64)|269|3095|477|6.5×|
> |2a3d(64)|360|2610|288|9.1×|
> |1d3z_ext(16)|508|831|94|8.9×|
> |mcl1_ext(16)|927|394|45|8.7×|
>
> **[W3. CSR index rebuild cost.]** Thanks for the suggestions. Our CSR rebuilding is designed to be efficient with bucket sorting, despite the neighbor list being updated every 1 step. We further provide isolated profiling of the CSR index rebuilding time as a percentage of total step time. As shown in Table R3, the CSR rebuild costs <1% of step time (as well as total time) across all the protein systems with various numbers of edges we profile on. This is because the bucket sort used for CSR construction is O(E) with a small constant, which is negligible vs the O(E×F) aggregation savings.
>
> **Table R3.**
> | Config | Edges | CSR rebuild time per step | % of step time |
> |---|---|---|---|
> | 1ENH (1 replica) | 30k | 0.15ms | 0.52% |
> | 1ENH (8 replicas) | 240k | 0.15ms | 0.52% |
> | 1ENH (64 replicas) | 1.9M | 0.28ms | 0.97% |
> | 2A3D (64 replicas) | 3.5M | 0.28ms | 0.97% |
> | MCL1 (16 replicas) | 2.0M | 0.27ms | 0.93% |
>
> **[Q1. CSR rebuild time.]** Please kindly see our response to W3.
>
> **[Q2. Single-trajectory vs. aggregate throughput.]** The reported ~1000 ns/day is the aggregate throughput across 64 parallel replicas of 1ENH protein systems. We further report single-trajectory throughput and throughput scaling table in terms of replica count, using the 1ENH protein.
> As shown in Table R4, the single-trajectory throughput of FlashSchNet is 1.4x faster than the baseline, and it scales up as we increase the batch size (e.g., 6.5x when the batch size is 64). Moreover, the near-constant memory footprint of FlashSchNet (e.g. 2.93 GB from bs=1 to bs=32) is a direct consequence of our IO-aware kernel design. Together, they demonstrate the superiority of scaling efficiency of our FlashSchNet and the new potentials that the memory saving unlocks.
>
> **Table R4.**
> | Batch Size | Flash (ts·mol/s) | Baseline (ts·mol/s) | Speedup | FlashSchNet Memory (GB) | Baseline Memory (GB) |
> |---|---|---|---|---|---|
> | 1 | 124.98 | 89.31 | 1.40× | 2.93 | 0.46 |
> | 4 | 502.72 | 311.22 | 1.62× | 2.94 | 6.46 |
> | 16 | 982.35 | 425.48 | 2.31× | 3.53 | 93.37 |
> | 32 | 2070.54 | 441.17 | 4.69× | 4.93 | 93.49 |
> | 64 | 3095.35 | 476.97 | 6.48× | 11.40 | 93.49 |
>
> **[Q3. NVE stability.]** Please kindly see our response to W1.
>
> **[Q4. Larger systems.]** Please kindly see our response to W2.
>
> We will further include the additional results and discussion of the above in our revised draft.

---

> > ### Author Rebuttal · Reviewer_NeN6 · 2026-04-03
> >
> > Thank you for the authors' detailed response. My concerns have been well addressed.

---

> > > ### Author Response · Authors · 2026-04-03
> > >
> > > Dear Reviewer NeN6,
> > >
> > > We are most grateful for your constructive questions and comments that surely help improve our draft. It’s also deeply appreciated that our response has been so kindly recognized with your increased score!
> > >
> > > Warmest regards,
> > >
> > > Authors of Submission #1626

---

### Decision · Program_Chairs · 2026-04-30

**Decision:**

Accept (regular)

**Comment:**

The authors present a sophisticated GPU implementation of a SchNet-style machine learned force field which is faster than classical force fields. This addresses a major issue with neural network force fields - they are typically more expensive to run than simpler classical MD force fields like MARTINI, AMBER, etc. The result is both an impressive piece of GPU engineering and a meaningful advance allowing for the best of both worlds - greater accuracy from the use of ML and neural networks with faster sampling from running on GPUs. The reviewers were unanimous that the work is of high quality and impactful. I recommend acceptance.